# Diversity of Cyanobacteria and Algae in Biological Soil Crusts of the Northern Ural Mountain Region Assessed through Morphological and Metabarcoding Approaches

**Elena Patova** [1,*]**, Irina Novakovskaya** [1] **, Evgeniy Gusev** [2] **and Nikita Martynenko** [2]

[1] Institute of Biology of Komi Science Centre of the Ural Branch of the Russian Academy of Sciences, Kommunisticheskaya, 28, Syktyvkar 167000, Russia; novakovskaya@ib.komisc.ru
[2] A.N. Severtsov Institute of Ecology and Evolution of the Russian Academy of Sciences, Leninsky Prospekt, 33, Moscow 119071, Russia; algogus@yandex.ru (E.G.); nikita-martynenko@yandex.ru (N.M.)
* Correspondence: patova@ib.komisc.ru

**Abstract:** In mountain regions, biological soil crusts (BSCs) provide essential ecological services by being involved in primary production and nitrogen fixation. Eukaryotic algae and cyanobacteria are important photosynthetic components of these unique cryptogamic communities. Here, we present an overview of the eukaryotic and prokaryotic diversity of such phototrophs in BSCs in the mountain tundra of the northern Ural Mountains. Such assessment is based on morphological surveys and the first metabarcoding analysis in the region. In total, 166 taxa of Cyanobacteria and 256 eukaryotic algae (including Euglenophyta, Ochrophyta, Dinophyta, Bacillariophyta, Chlorophyta, and Charophyta) were identified. For the first time, 86 taxa new to the BSCs of the high-mountain belt of the region were discovered. Considering species composition, Cyanobacteria and Chlorophyta are the most abundant taxa in all the analyzed BSCs. The genera *Nostoc, Coccomyxa, Chlamydomonas, Leptolyngbya, Stenomitos, Pycnacronema, Stigonema,* and *Eunotia* had the highest number of taxonomic units. These groups shape the structure, function, and ecology of the BSC communities in the studied region. Our results show that BSCs in the tundras of the Ural Mountains have a high active and passive biodiversity of terrestrial cyanobacteria and algae. Both implemented methods resulted in similar results with a comparable number of algae and cyanobacteria species per sample. Metabarcoding could be implemented in future in the region to accurately screen photosynthetic organisms in BSCs.

**Keywords:** biological soil crusts; terrestrial cyanobacteria and eukaryotic algae; mountain tundra; Polar; Subpolar and Northern Urals

## 1. Introduction

Biological soil crusts (BSCs) are primal producers of organic matter in mountain and arctic regions' soils [1–4]. Microalgae and cyanobacteria are important components of these cryptogamic communities, playing a critical role in nutrient and nitrogen cycling and highly influencing soil oxygen and hydrological regimes. They are active in the top soil layer, forming a thin crust that helps to stabilize soil, maintain its structure, and protect against wind, thermokarst, and water erosions [1–6]. In extreme habitats such as alpine and polar ecosystems, where soils are often shallow and exposed, BSCs dominate, providing an important source of nutrition and water for small organisms that would otherwise struggle to survive. Despite the high resilience to extreme conditions of these unique communities, climate change will modify the habitats of BSCs through shifts in temperature, irradiance, and precipitation, all of which will result in a significant restructuring of BSC communities and their associated diversity and composition [7,8]. Such alternations in a leading component of the vegetation cover in mountain regions will likely trigger drastic changes in soil formation, the replacement of species, and the alteration of nutrient cycles. The taxonomic diversity and species composition of BSCs are highly sensitive to the

smallest fluctuations occurring in the ecosystem, thus serving as a good indicator of the effect climate change has on polar and high-mountain ecosystems. To monitor changes in BSCs under climatic fluctuations, extensive data on their diversity, structure, and functional characteristics are required, covering a wide range of habitats, including high-mountain, arctic, and polar ecosystems. Such studies have been actively carried out on Svalbard and Antarctica [7–18] but are still rare for the Russian Arctic, even though its territories are vast and diverse [19–27]. Traditionally, the diversity of cyanobacteria and soil algae in BSCs is recovered through strain isolation on selective media, with further identification of species based on morphological characters. Only a fraction of the total diversity of algae and cyanobacteria can be revealed through such an approach. Modern advances in taxonomic tools have introduced DNA metabarcoding with the ability to recover hidden diversity. This method has been proven to be successful in soil samples, revealing interesting, unique, and cryptic species and benefiting from identifying multiple taxa simultaneously in a single sample from total DNA [28]. In addition, metabarcoding enables complex studies of whole communities with the potential to detect ecological and functional networks and assess differences in species richness between similar habitats [29]. DNA metabarcoding has significantly expanded the knowledge of microorganism communities (algae, bacteria, fungi, and archaea) in a wide range of natural biomes [18]. Although this method is rarely implemented in BSCs in the Arctic and mountain regions, it is slowly starting to gain popularity, with a handful of articles published with promising and interesting results, in particular in the Arctic and Antarctica [18] and in the French Alps [30]. Metabarcoding studies have revealed that BSCs in the Arctic and Antarctica form unique ecosystems with an unexpectedly high biodiversity of cyanobacteria and eukaryotic algae, which were found to be the most abundant among autotrophic organisms in the BSCs [18]. A similar pattern was also observed in the French Alps [30]. Further use of DNA metabarcoding in studies of BSCs should aim to expand their application in diverse ecosystems under vast ecological conditions, enlarging their spatial coverage.

BSCs are widespread on bare spots in mountain tundra and in open forests, common at high altitudes in the Ural mountains. This mountain chain is located in northern Eurasia, and the Polar, Subpolar, and Northern Urals are its highest regions. The Urals are known for a remarkable diversity of flora and fauna, which is attributed to its unique location between Europe and Asia. The last glaciation also had a profound impact and shaped its present-day diversity. All of this makes the region particularly interesting for diversity studies.

The diversity of algae and cyanobacteria in BSCs has been studied since 2005 [20–26]. Knowledge of the species richness, abundance, and functional characteristics of BSC algae was accumulated through mostly traditional methods. Data on the diversity of cyanobacteria are actively expanding through ongoing research efforts focused on previously inaccessible areas and with the implementation of novel and advanced molecular techniques [20–27].

Combining morphological studies and DNA metabarcoding, the current paper summarizes data on the taxonomic and structural diversity of cyanobacteria and eukaryotic algae in biological soil crust (BSC) communities in mountain and tundra habitats in the northern regions of the Urals.

## 2. Materials and Methods

### 2.1. BSCs Sampling Sites

The study combined a literature review and new data to make an inventory of algal and cyanobacterial species in BSCs in three regions of the Ural Mountains, located in the northern Russian Arctic (Eurasia) [20–27,31]. The sampling locations are presented in Table 1 and Figure 1. The Polar, Subpolar, and Northern Urals lay in a zone of harsh continental climate, characterized by long cold winters, cool summers, short growing seasons, relatively high precipitation, and extremely low evaporation [32]. The average annual air temperature is −4.4 °C, with January as the coldest month (−18–21 °C) and July (8–11 °C) as the warmest. In June-July, temperatures can exceed +20–30 °C, though summers in the Urals are generally cold with unstable weather. The growing season lasts

less than 80 days. The windward (western and southwestern) slopes of the Urals experience the largest rainfall due to the natural movement of the air masses from the Atlantic, and thus, they are the most humid (1000–1500 mm) [32]. The average temperatures in January and July and the annual precipitation in the Ural regions are as follows: Polar (−22 °C, +10–11 °C, 1000 mm), Subpolar (−22 °C, +12–13 °C, 1100 mm), and Northern (−21 °C, +15–17 °C, 1200 mm) [32]. The studied BSCs are located in bare spots on acidic soils with low base saturation and low nitrogen, phosphorus, and other biogenic element content (Table 2). The soil moisture of bare spots ranges from 2.2 to 50% [22–24,26,27].

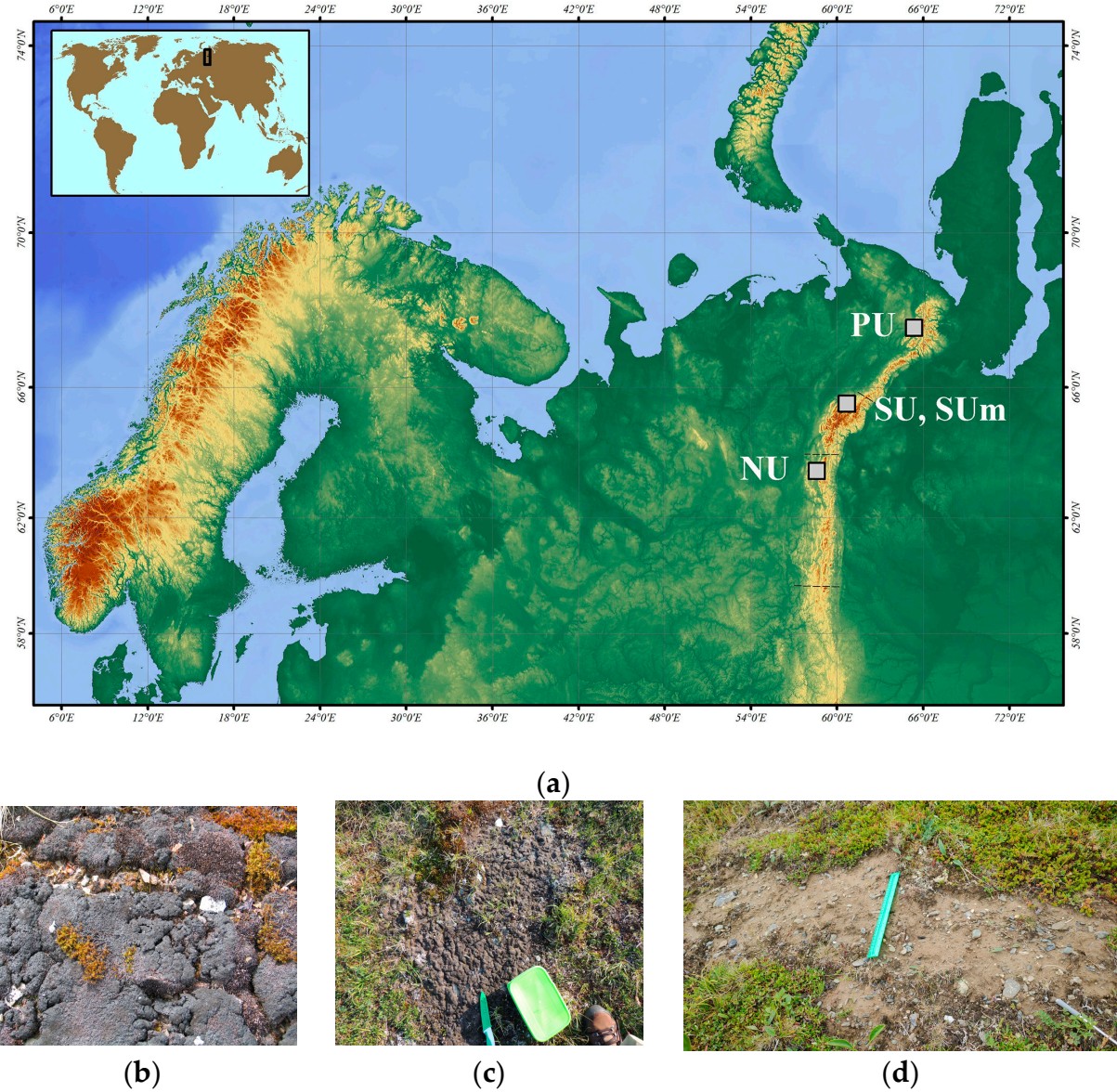

**Figure 1.** BSCs samples and locations: (**a**)—map of the studied area (ArcGIS 9.2 OSM Open-TopoMap.org); (**b**–**d**)—images of the sampled BSCs. A summary of the sample plots is outlined in Table 1.

**Table 1.** BSCs sampling locations in the Ural Mountains.

| Region | Collection Date | Sampling Site | Elevation (m) | Coordinates |
|---|---|---|---|---|
| | August, 2011 | Mount Konstantinov Kamen | 483 | 68.48 N, 66.23 E |
| | August, 2011 | Malyi Manisey | 525 | 68.47 N, 66.34 E |
| Polar Urals (PU) | July 2019, 2022 | Ochenyrd ridge | 153–1161 | 67.91–68.19 N, 65.21–66.05 E |
| | August 2003, 2013 | Chrebtovy Ridge | 153 | 67.38 N, 64.67 E |
| Subpolar Urals (SU) and metabarcoding samples (SUm) * | August 2005, 2009; July, 2010, 2012, 2019, 2021 * | Maldynyrd Ridge | 1538 | 65.22–65.31 N, 60.26–60.52 E |
| | August, 2005, 2009; July, 2010, 2012, 2019, 2021 * | Barkova Mountain | 1366 | 65.21 N, 60.26 E |
| | July, 2010 | Starik Mountain | 1233 | 65.16 N, 60.29 E |
| Northern Urals (NU) | June–July, 2018–2019 | Telpos-iz Ridge | 495–1248 | 63.40–63.54 N, 58.54–59.07 E |
| | July, 2016, 2018 | Mount Pelener | 585–1067 | 63.38 N, 58.90 E |

Note: *—samples for metabarcoding.

**Table 2.** Environmental parameters of the soil underneath the sampled BSCs.

| Parameter | PU | SU, SUm | NU |
|---|---|---|---|
| pH ($H_2O$) | 3.3–6.9 | 4.08–6.2 | 4.38–5.48 |
| pH (KCl) | 3.2–6.1 | 2.7–5.3 | 3.40–4.23 |
| C [mg/kg] | 0.42–13.9 | 0.22–3.9 | 0.35–5.8 |
| N [mg/kg] | 0.042–0.97 | 0.026–0.27 | 0.035–0.51 |
| $P_2O_5$ [mg/kg] | 30.90–225.6 | 2.6–1520 | 15–500 |
| $Ca^{2+}$ [mg/kg] | 0.71–73.71 | 0.26–72 | <0.5–2.79 |
| $Mg^{2+}$ [mg/kg] | 0.26–2.50 | 0.12–24 | <0.10–0.64 |

Note: The range of values is given for the parameters according to [22].

### 2.2. BSCs Sampling and Algae Cultivation

Each sample consisted of 5–6 subsamples with 3 cm × 5 cm × 2 cm dimensions. Samples were cut from the soil with a knife (flame sterilized), placed in paper bags, air-dried, and transported to the laboratory [33]. Algae and cyanobacteria diversity were revealed by studying freshly collected samples under a microscope (algae and cyanobacteria forming mass overgrowth on the soil surface were identified) and analyzing mixed/monocultures grown on liquid or solid BG11 media [33]. The protocol used to isolate and grow the strains was adapted from Andersen [33]. The cultivating conditions used for mixed and unialgal cultures were as follows: 45 μmol m$^{-2}$s$^{-1}$ PAR under Uniel ULI-P11-35W/SPFR IP40 WHITE l phytolamp (Beijing, China) with 12/12 h light/dark period at 25 °C. Additional details about the methods used to study the diversity of algae and cyanobacteria in the Ural region can be found in published works [21–27]. The strains of the algae and cyanobacteria described in the article were deposited in the culture collection of the Institute of Biology, Komi Scientific Center, Ural Branch of the Russian Academy of Sciences (SYKOA, Syktyvkar, Russia, https://ib.komisc.ru/sykoa/eng/, accessed on 8 August 2023).

### 2.3. Algae and Cyanobacteria Identification

The species were visualized and identified under a Nikon 80i Nomarski DIC Microscope (Tokyo, Japan), equipped with Nikon Digital Sight Ds—2Mv camera (Tokyo, Japan). Detailed morphological descriptions and pictures for some taxa in the checklist are available at https://ib.komisc.ru/sykoa/eng/ and http://kpabg.ru/cyanopro. The algal and cyanobacteria species were described following the guidelines of Russian and

international identification keys [34–39] and modern systematic reviews for individual taxa. The molecular analysis was performed for some strains of algae and cyanobacteria analyzed in this work and published [40–42].

The current nomenclature for algae and cyanobacteria is devised according to international database "AlgaeBase" [43].

### 2.4. DNA Extraction

For metabarcoding, 8 samples were taken from cryogenic spots in different mountain tundra communities in the Subpolar Urals in July 2021 (Table 1, Figure 1). The BSCs were cut into cubes ($3 \times 5 \times 2$ cm) in three replicates, placed in sterile Petri dishes, and immediately frozen at $-45$ °C. Before total DNA isolation, each sample was thawed, and 0.25 g of a mixed BSC sample from three replicates was taken. The total DNA was isolated using the DNeasy PowerSoil Kit according to the manufacturer's protocol.

For eukaryotic algae, PCR amplifications targeted the SSU V9 region and ITS1. The primer pair V918S BoenF (5′–GTACACACCGCCCGTC–3′)—ITS2_broad (5′–GCTGCGTTC TTCATCGWTR–3′) was chosen, as proposed by Boenigk et al. [44] and Bock et al. [45]. These primers cover the variable V9 region of the 18S rDNA and complete ITS1, which allows them to be used both for a phylogenetic analysis of the 18S rDNA region and for the species identification of algae. Currently, for most algae groups, ITS data are required for species identification. For cyanobacteria, 16S1515F (5′–AGTCGTAACAAGGTAGCCGTA CC–3′)—B23SR (5′–CTTCGCCTCTGTGTGCCTAGG–3′) was used, spanning the end of the 16S–ITS-beginning of 23S rDNA [46,47]. DNA samples were transferred to the «Evrogen» (Moscow, Russia) for subsequent sequencing.

### 2.5. Amplification, Library Preparation, and Sequencing

The quality of the obtained genomic DNA samples was checked on an agarose gel. Amplification of variable regions was carried out using specific primers. Sequencing was performed on Illumina MiSeq, set v3, pair end, read length $2 \times 300$ cycles. Libraries were prepared for sequencing according to the protocol described in the 16S Metagenomic Sequencing Library Preparation manual (Part # 15,044,223 Rev. B; Illumina). After receiving the amplicons, the libraries were purified and mixed using the SequalPrep™ Normalization Plate Kit (ThermoFisher, Cat # A10510–01). A quality control of the obtained pools of libraries was carried out using the Fragment Analyzer system, and a quantitative analysis was carried out using qPCR.

Each library was sequenced on Illumina MiSeq (read length—300 bp on both sides of the fragments) using a MiSeq Reagent Kit v3 (600 cycles). FASTQ files were generated using bcl2fastq v2.17.1.14 Conversion Software (Illumina, San Diego, USA). After filtering the data, 834,606 paired reads were obtained for 8 samples, sequenced with eukaryotic primers. For cyanobacteria, 1092254 paired reads were obtained for 8 samples. The paired reads were obtained for 8 samples. The PhiX phage library was used to control sequencing parameters. Samples were demultiplexed by the sequencing company using Illumina sequencing software (version v3).

### 2.6. Bioinformatic Pipeline

The base quality of the sequence reads was checked using the FastQC software [48]. The sequence filtering was performed using QIIME2 tools [49]. For filtering, denoising, dereplication, and obtaining amplicon sequence variants, the DADA2 plugin for QIIME2 was used (max_ee = 2; trunc_q = 2; pooling method = independent; chimera method = consensus). In this work, we used only forward sequences with a length of 264 bp for cyanobacteria and 259 bp for eukaryotes. Operational taxonomic units (OTUs) were delineated at a similarity of 97% for eukaryotic and 93% for prokaryotic datasets using the VSEARCH plugin for QIIME2 [49,50]. Taxonomic affiliation was determined using the GenBank database through the BLASTn algorithm to assign them to species and genus levels. For this, the SEED2 program was used [51]. For the species identification of eukaryotes, a threshold

of similarity of 97% was chosen for the V918S–ITS1 sequences. In the case of presence of only 18S sequences in the GenBank database, a threshold of 99% was used. Sequences with lower values were identified at the genus level.

To identify cyanobacteria, a threshold of 93% similarity was chosen for ITS sequences, as proposed by many authors [52–55]. Sequences with lower values were identified only at a genus rank. All identifications at the species and genus level have been checked on maximum likelihood (ML) phylogenetic trees. For this, sequences have been aligned using MAFFT [56] and processed in RAML [57]. In this work, the analysis used only data on OTUs that were identified by their genus and species clades for a correct comparison with morphological data. OTUs with unclear systematic positions have not yet been considered due to the possible errors and incorrect interpretations of data that could arise due to incomplete reference databases and unclear nutritional modes of unknown organisms.

The sequences of the OTUs of cyanobacteria and eukaryotic algae are represented in the Supplementary File S1. The datasets analyzed during the current study are available in the NCBI repository (accession: PRJNA1016378).

## 3. Results

### 3.1. Diversity of Algae and Cyanobacteria in BSCs Based on Morphological Analysis

Based on the morphological analysis, 343 taxa (below the genus level) of soil algae and cyanobacteria from six phyla were discovered in BSCs from various mountain tundra communities in the northern Urals. The phyla were Cyanobacteria (135), Chlorophyta (131), Bacillariophyta (52), Charophyta (14), Ohrophyta (12), and Euglenophyta (1) (Table 3).

**Table 3.** List of terrestrial cyanobacteria and eukaryotic algae species in BSCs in diverse mountain tundra communities in the northern regions of the Urals: morphological data for PU (Polar Urals), SU (Subpolar Urals), and NU (Northern Urals) and metabarcoding data for SUm (Subpolar Urals). N—number of OTUs.

| Taxon | PU | SU | NU | SUm |
|---|---|---|---|---|
| **Cyanobacteria** | | | | |
| *Albertania* spp. | | | | + (N = 2) |
| *Aliterella* sp. | | | | + (N = 1) |
| *Ammatoidea normanii* West et G.S.West | | | + | |
| *Anabaena cylindrica* Lemmerm.[1] | | + | | |
| *Anabaena* spp. | | + | + | + (N = 1) |
| *Ancylothrix* sp. | | | | + (N = 1) |
| *Aphanocapsa fuscolutea* Hansg. | + | | | |
| *Aphanocapsa muscicola* (Menegh.) Wille | + | + | + | |
| *Aphanocapsa parietina* Nägeli | + | + | | |
| *Aphanocapsa rivularis* (Carmich.) Rabenh. | + | | | |
| *Aphanocapsa* spp. | + | + | + | |
| *Aphanothece castagnei* (Kütz.) Rabenh. | | + | | |
| *Aphanothece microscopica* Nägeli | | + | | |
| *Aphanothece saxicola* Nägeli | + | + | + | |
| *Aphanothece stagnina* (Spreng.) A.Braun | + | + | | |
| *Aphanothece pallida* (Kütz.) Rabenh. | + | + | | |
| *Aphanothece* sp. | + | | | |
| *Calothrix braunii* Bornet et Flahault | | + | + | |
| *Calothrix clavata* G.S.West | | + | | |
| *Calothrix elenkinii* Kossinsk. [1] | | + | | |
| *Calothrix parietina* Thur. ex Bornet et Flahault | + | + | + | |
| *Calothrix* spp. | + | + | + | |
| *Chamaesiphon polonicus* (Rostaf.) Hansg. | + | | | |
| *Chlorogloea purpurea* Geitler | + | | | |
| *Chroococcidiopsis* sp. | | | | + (N = 1) |
| *Chroococcus cohaerens* (Bréb.) Nägeli | + | + | | |
| *Chroococcus giganteus* W.West | | + | | |
| *Chroococcus minor* (Kütz.) Nägeli | | + | | |
| *Chroococcus minutus* (Kütz.) Nägeli | + | + | | |
| *Chroococcus spelaeus* Erceg. | + | | | |

**Table 3.** *Cont.*

| Taxon | PU | SU | NU | SUm |
|---|---|---|---|---|
| **Cyanobacteria** | | | | |
| *Chroococcus tenax* (Kirchn.) Hieronymus | + | | | |
| *Chroococcus turgidus* (Kütz.) Nägeli | | + | | |
| *Chroococcus varius* A.Braun | + | | | |
| *Chroococcus* spp. | + | | | + (N = 2) |
| *Cyanobacterium cedrorum* (Sauv.) Komárek, Kopecky et Cepák | | + | | |
| *Cyanosarcina chroococcoides* (Gaitler) Kováčik | + | | | |
| *Cyanosarcina* sp. | + | | | |
| *Cyanothece aeruginosa* (Nägeli) Komärek | + | | | |
| *Cyanothece* spp. | | | | + (N = 2) |
| *Cylindrospermum muscicola* Kutz. ex Bornet et Flahault | | | + | |
| *Cylindrospermum* spp. | + | | | + (N = 2) |
| *Dactylothamnos antarcticus* Fiore, Genuario, Komárek et al. | | | | + (N = 1) |
| *Dasygloea* cf. *lamyi* (Gomont ex Gomont) Senna et Komárek [1] | | + | | |
| *Dasygloea* sp. | + | | | |
| *Desmonostoc muscorum* (Bornet et Flahault) Hrouzek et Ventura [1] | + | + | + | |
| *Desmonostoc* spp. | | | | + (N = 5) |
| *Dichothrix gypsophila* Bornet et Flahault | + | + | | |
| *Dolichospermum* spp. | | | | + (N = 3) |
| *Drouetiella lurida* (Gomont) Mai, J.R.Johansen et Pietrasiak [1,2] [41] | | + | | |
| *Drouetiella ramosa* Davydov, Vilnet, Patova et Novakovskaya [1,2] [58] | | + | | |
| *Drouetiella* spp. | | | | + (N = 4) |
| *Eucapsis minor* (Skuja) Elenkin | | + | | |
| *Fischerella ambigua* (Kütz. ex Bornet et Flahault) Gomont f. *majuscula* (Woron.) Elenkin | | + | | |
| *Fischerella major* Gomont | | + | | |
| *Fischerella muscicola* Gomont [1] | + | + | + | |
| *Fischerella* spp. | | + | + | |
| *Geitlerinema* sp. | | | | + (N = 1) |
| *Gloeocapsa alpina* Nägeli | + | + | | |
| *Gloeocapsa compacta* Kütz. | + | + | | |
| *Gloeocapsa kuetzingiana* Nägeli ex Kütz. | + | | | |
| *Gloeocapsa punctata* Nägeli | | + | | |
| *Gloeocapsa ralfsii* (Harvey) Lemmerm. | + | | | |
| *Gloeocapsa rupestris* Kütz. | + | + | | |
| *Gloeocapsa sanguinea* (C.Agardh) Kütz. | + | | | |
| *Gloeocapsa violacea* Kütz. | + | + | | |
| *Gloeocapsopsis dvorakii* (Novácek) Komárek et Anagn. ex Komárek | | + | + | |
| *Gloeocapsopsis magma* (Bréb.) Komárek et Anagn. ex Komárek | + | + | + | |
| *Gloeocapsopsis* sp. | + | | | |
| *Gloeothece confluens* Nägeli | + | + | | |
| *Gloeothece rupestris* (Lyngb.) Bornet | + | + | | |
| *Gloeothece tepidariorum* (A.Braun) Lagerh. | + | | | |
| *Haloleptolyngbya* sp. | | | | + (N = 1) |
| *Hapalosiphon pumilus* Kirchner ex Bornet et Flahault | | + | | |
| *Heteroscytonema crispum* (Bornet ex De Toni) G.B.McGregor et Sendall | | + | | |
| *Jaaginema pseudogeminatum* (G.Schmid) Anagn. et Komárek | | + | | |
| *Kamptonema* sp. | | | | + (N = 1) |
| *Kovacikia* sp. | | | | + (N = 1) |
| *Leptolyngbya angustissima* (W.West et G.S.West) Anagn. et Komárek | | + | | |
| *Leptolyngbya boryana* (Gomont) Anagn. et Komárek | | | + | |
| *Leptolyngbya foveolarum* (Gomont) Anagn. et Komárek | + | + | + | |
| *Leptolyngbya gracillima* (Hansg.) Anagn. et Komärek | + | | | |
| *Leptolyngbya nostocorum* (Bornet ex Gomont) Anagn. et Komárek | | + | | |
| *Leptolyngbya notata* (Schmidle) Anagn. et Komárek | | + | + | |
| *Leptolyngbya sieminskae* D.Richter et Matula | + | | | |
| *Leptolyngbya subtilissima* (Hansg.) Komárek | | | + | |
| *Leptolyngbya tenuis* (Gomont) Anagn. et Komárek | | + | | |
| *Leptolyngbya* spp. [1] | + | + | + | + (N = 6) |
| *Lyngbya* sp. | + | | | |
| *Microchaete tenera* Thur. ex Bornet et Flahault | | + | | |
| *Microcoleus autumnalis* (Gomont) Strunecky, Komarek et J.R.Johans. [1] | + | + | + | |
| *Microcoleus* cf. *lacustris* Farlow ex Gomont | | | + | |
| *Microcoleus fonticola* (Kirchner ex Hansg.) Strunecky, Komárek et J.R.Johans. | | + | | |
| *Microcoleus paludosus* Gomont | + | + | + | |
| *Microcoleus vaginatus* Gomont | + | + | | + (N = 1) |
| *Microcoleus* spp. | | + | + | |
| *Microcystis* sp. | | + | | |
| *Nodosilinea* sp. | | | | + (N = 1) |

**Table 3.** *Cont.*

| Taxon | PU | SU | NU | SUm |
|---|---|---|---|---|
| **Cyanobacteria** | | | | |
| *Nostoc commune* Vaucher ex Bornet et Flahault f. *ulvaceum* Elenkin [1,2] | + | + | | |
| *Nostoc commune* Vaucher ex Bornet et Flahault [1,2] | + | | + | + (N = 1) |
| *Nostoc edaphicum* N.V.Kondrateva | + | + | + | + (N = 1) |
| *Nostoc linckia* (Roth) Bornet et Flahault [1,2] | + | | | |
| *Nostoc microscopicum* Carmich. ex Bornet et Flahault | + | | | |
| *Nostoc paludosum* Kütz. ex Bornet et Flahault | + | + | | |
| *Nostoc punctiforme* Har. [1,2] | + | + | + | |
| *Nostoc* spp. | + | | + | + (N = 30) |
| *Nostoc* sp. (photobiont of liver mosses) [1] | | | + | |
| *Oculatella crustae-formantes* P.Jung, Briegel-Williams, Mikhailyuk et Büdel | | | | + (N = 1) |
| *Oculatella* spp. | | | | + (N = 3) |
| *Oscillatoria tenuis* C.Agardh ex Gomont | + | | | |
| *Petalonema densum* (Bornet ex Bornet et Flahault) Migula | | + | | |
| *Petalonema incrustans* Komárek | + | | | |
| *Phormidesmis mollis* (Gomont) Turicchia, Ventura, Komárková et Komárek [1] | | + | | |
| *Phormidesmis nigrescens* (Komarek) Raabová, L.Kovacik, Elster et Strunecký | | | | + (N = 1) |
| *Phormidesmis* sp. | + | | | |
| *Phormidium ambiguum* Gomont [1] | + | + | | |
| *Phormidium corium* Gomont [1] | + | + | + | |
| *Phormidium interruptum* Kütz. ex Forti | | + | | |
| *Phormidium kuetzingianum* (Kirchn. ex Hansg.) Anagn. et Komárek | + | + | | |
| *Phormidium puteale* (Montagne ex Gomont) Anagn. et Komárek [1] | + | | + | |
| *Phormidium* spp. | + | + | + | + (N = 2) |
| *Pleurocapsa aurantiaca* Geitler | | | + | |
| *Porphyrosiphon fuscus* Gomont ex Frémy [1] | | + | | |
| *Porphyrosiphon lomniczensis* (Kol) Anagn. et Komárek [1] | | + | | |
| *Porphyrosiphon* sp. | | | | + (N = 1) |
| *Potamolinea aerugineocaerulea* (Gomont) M.D.Martins et Branco [1] | + | + | + | |
| *Pseudanabaena* spp. | + | | | + (N = 2) |
| *Pseudophormidium hollerbachianum* (Elenkin) Anagn. [1] | | + | | |
| *Pycnacronema* spp. | | | | + (N = 11) |
| *Schizothrix calcicola* Gomont | + | | | |
| *Schizothrix lardacea* Gomont | + | | | |
| *Schizothrix fuscescens* Kutz. ex Gomont | | + | | |
| *Schizothrix* spp. | + | + | + | |
| *Scytonema hoffmannii* C.Agardh ex Bornet et Flahault [1] | + | + | + | |
| *Scytonema hyalinum* N.L.Gardner | | | | + (N = 1) |
| *Scytonema ocellatum* Lyngbye ex Bornet et Flahault | + | + | | |
| *Scytonema* spp. | + | + | + | |
| *Scytonematopsis crustacea* (Thur. ex Bornet et Flahault) Koválik et Komárek [1,2] | | + | | |
| *Shackletoniella* spp. | | | | + (N = 2) |
| *Snowella* sp. | | | | + (N = 1) |
| *Stenomitos frigidus* (F.E.Fritsch) Miscoe et J.R.Johans. | + | + | + | |
| *Stenomitos hiloensis* Johansen, Gargas et Shalygin | | | | + (N = 1) |
| *Stenomitos kolaensis* Shalygin, Shalygina et Johansen | | | | + (N = 1) |
| *Stenomitos* spp. [1,2] [41] | + | | | + (N = 10) |
| *Stigonema hormoides* Bornet et Flahault | + | + | | |
| *Stigonema informe* Kütz. ex Bornet et Flahault [1,2] | + | + | | |
| *Stigonema mamillosum* C.Agardh ex Bornet et Flahault | + | + | + | |
| *Stigonema minutum* Hassall ex Bornet et Flahault | + | + | + | |
| *Stigonema ocellatum* Thur. ex Bornet et Flahault [1,2] | + | + | + | |
| *Stigonema* spp. | + | + | + | + (N = 10) |
| *Symplocastrum friesii* (Gomont) Kirchn. [1] | + | + | + | |
| *Synechococcus elongatus* (Nägeli) Nägeli | | + | + | |
| *Synechocystis crassa* Woron. | | + | | |
| *Synechocystis fuscopigmentosa* Kovácik | | | | + (N = 1) |
| *Synechocystis* spp. | | | | + (N = 2) |
| *Tenebriella curviceps* (C.Agardh ex Gomont) Hauerová, Hauer et Kaštovský | | | | + (N = 1) |
| *Tildeniella alaskaensis* Strunecky, Raabova, Bernardova, A.P.Ivanova, Semanova, Crossley et Kaftan | | | | + (N = 1) |
| *Tildeniella* sp. | | | | + (N = 1) |
| *Timaviella* spp. | | | | + (N = 8) |
| *Tolypothrix distorta* Kütz. ex Bornet et Flahault | + | + | | + (N = 1) |
| *Tolypothrix lanata* Wartm. ex Bornet et Flahault | | + | | |
| *Tolypothrix saviczii* Kossinsk. | | + | | |
| *Tolypothrix tenuis* Kütz. ex Bornet et Flahault [1] | + | + | + | |
| *Tolypothrix fasciculata* Gomont [1] | | + | | |
| *Tolypothrix* spp. | + | + | | + (N = 3) |

**Table 3.** *Cont.*

| Taxon | PU | SU | NU | SUm |
|---|---|---|---|---|
| **Cyanobacteria** | | | | |
| *Trichormus variabilis* (Kütz. ex Bornet et Flahault) Komárek et Anagn. | | + | | |
| *Wilmottia* sp. | | | | + (N = 1) |
| Total | 83 | 94 | 44 | 221 OTUs Identified—135 Not identified— 86 |
| **Dinophyta** | | | | |
| *Nusuttodinium* sp. | | | | + (N = 1) |
| *Peridinium* sp. | | | | + (N = 1) |
| Total | 0 | 0 | 0 | 8 OTUs Identified—2 Not identified—6 |
| **Euglenophyta** | | | | |
| *Astasia* sp. | | | + | |
| Total | 0 | 0 | 1 | 3 OTUs Identified—0 Not identified—3 |
| **Cryptophyta** | | | | |
| Total | | | | 3 OTUs Identified—0 Not identified—3 |
| **Ochrophyta/Chrysophyceae** | | | | |
| *Chromulina chionophilia* Stein | | | | + (N = 1) |
| *Chromulina* spp. | | | | + (N = 3) |
| *Chrysocapsa vernalis* Starmach | | | | + (N = 1) |
| *Kremastochrysopsis austriaca* Remias, Procházková et R.A.Andersen | | | | + (N = 1) |
| *Spumella* spp. | | | | + (N = 4) |
| Total | 0 | 0 | 0 | 19 OTUs Identified—10 Not identified—9 |
| **Ochrophyta/Xanthophyceae** | | | | |
| *Botrydiopsis eriensis* J.W.Snow [1] | | + | + | |
| *Bumilleria sicula* Borzi [1] | | + | | |
| cf. *Chlorobotrys simplex* Pascher | | + | | |
| *Chlorobotrys* sp. | | | | + (N = 1) |
| *Characiopsiella minima* (Pascher) R.Amaral, K.P.Fawley, Nemcová, T.Sevcíková, Lukesová, M.W.Fawley, L.M.A.Santos et M.Eliás | | + | | |
| *Characiopsis minutissima* Pascher [1] | | | + | |
| *Monodopsis subterranea* (J.B.Petersen) D.J.Hibberd | | | | + (N = 1) |
| *Pleurochloris pyrenoidosa* Pascher | | + | | |
| *Tribonema vulgare* Pascher [1] | + | | | |
| *Tribonema* sp. | | + | | |
| *Vischeria helvetica* (Vischer et Pascher) D.J.Hibberd [1] | + | + | | |
| *Vischeria magna* (Petersen) Kryvenda, Rybalka, Wolf et Friedl [1,2] | + | + | + | + (N = 1) |
| *Vischeria stellata* (Chodat) Pascher | | | | + (N = 1) |
| *Vischeria* sp. | | | + | |
| *Xanthonema bristolianum* (Pascher) P.C.Silva [1] | | | + | |
| Total | 3 | 8 | 5 | 4 OTUs Identified—4 Not identified—0 |
| **Bacillariophyta** | | | | |
| *Achnanthidium lineare* W.Smith | | + | | |
| *Achnanthidium minutissimum* (Kütz.) Czarnecki | | + | | |
| *Aulacoseira italica* (Ehrenb.) Simonsen | | + | | |
| *Caloneis aerophila* W.Bock | | + | | |
| *Cavinula* cf. *lapidosa* (Krasske) Lange-Bert. | | + | | |
| *Chamaepinnularia begeri* (Krasske) Lange-Bert. | | + | | |
| *Chamaepinnularia soehrensis* (Krasske) Lange-Bert. et Krammer | | + | | |
| *Diatoma tenuis* C.Agardh | | + | | |
| *Discostella* sp. | | | | + (N = 1) |
| *Encyonema gracile* Rabenh. | | + | | |
| *Encyonema minutum* (Hilse) D.G.Mann | | + | | |
| *Eunotia bilunaris* (Ehrenb.) Schaarschm. | | + | | |
| *Eunotia diodon* Ehrenb. | | + | | |
| *Eunotia fallax* A.Cleve | | + | | |
| *Eunotia incisa* W.Smith ex W.Gregory | | + | | |
| *Eunotia intermedia* (Krasske ex Hustedt) Nörpel et Lange-Bert. | | + | | |
| *Eunotia microcephala* Krasske | | + | | |
| *Eunotia paludosa* Grunow | | + | | |
| *Eunotia praerupta* Ehrenb. | | + | | |
| *Eunotia septentrionalis* Østrup | | + | | |
| *Eunotia* sp. | | | + | |

**Table 3.** *Cont.*

| Taxon | PU | SU | NU | SUm |
|---|---|---|---|---|
| **Bacillariophyta** | | | | |
| *Fragilaria radians* (Kütz.) D.M.Williams et Round | | + | | |
| *Fragilaria vaucheriae* (Kütz.) J.B.Petersen | | + | | |
| *Fragilaria* sp. | | + | | |
| *Gomphonema angustatum* (Kütz.) Rabenh. | | + | | |
| *Gomphonema brebissonii* Kütz. | | + | | |
| *Hannaea arcus* (Ehrenb.) R.M.Patrick | | + | | |
| *Hantzschia amphioxys* (Ehrenb.) Grunow | + | + | + | |
| *Hantzschia amphioxys* (Ehrenb.) Grunow f. *capitata* O.Müll. | | | + | |
| *Microcostatus krasskei* (Hustedt) J.R.Johans. et Sray | | + | | |
| *Navicula* spp. | + | + | + | |
| *Neidium alpinum* Hustedt | | + | | |
| *Neidium bisulcatum* (Lagerst.) Cleve | | + | | |
| *Nitzschia frustulum* (Kütz.) Grunow | | + | | |
| *Nitzschia palea* (Kütz.) W.Smith | | + | | |
| *Nitzschia perminuta* Grunow | | + | | |
| *Pinnularia appendiculata* (C.Agardh) Schaarschm. | | + | | |
| *Pinnularia borealis* Ehrenb. | | + | + | |
| *Pinnularia* cf. *bullacostae* Krammer et Lange-Bert. | | + | | |
| *Pinnularia* cf. *microstauron* (Ehrenb.) Cleve var. *rostrata* Krammer | | + | | |
| *Pinnularia streptoraphe* Cleve | | + | | |
| *Pinnularia subcapitata* W.Gregory | | + | | |
| *Pinnularia subrostrata* (A.Cleve) A.Cleve | | + | | |
| *Pinnularia* spp. | + | + | + | |
| *Psammothidium helveticum* (Hust.) Bukht. et Round | | + | | |
| *Psammothidium kryophilum* (J.B.Petersen) E.Reichardt | | + | | |
| *Sellaphora* spp. | | | | + (N = 3) |
| *Stauroneis agrestis* Petersen | + | + | | |
| *Stauroneis anceps* Ehrenb. | | + | | |
| *Staurosira subsalina* (Hustedt) Lange-Bert. | | + | | |
| *Staurosirella pinnata* (Ehrenb.) D.M.Williams et Round | | + | | |
| *Tabellaria flocculosa* (Roth) Kütz. | | + | | |
| *Tabellaria* sp. | + | | | |
| *Thalassiosira pseudonana* Hasle et Heimdal | | | | + (N = 1) |
| *Ulnaria ulna* (Nitzsch) Compère | | + | | |
| Total | 5 | 49 | 6 | 11 OTUs Identified—5 Not identified—6 |
| **Chlorophyta** | | | | |
| *Apatococcus lobatus* (Chodat) J.B.Petersen | | | | + (N = 1) |
| *Asterochloris excentrica* (Archibald) Skaloud et Peksa [1] | + | | + | |
| *Asterochloris italiana* (Archibald) Skaloud et Peksa | | | | + (N = 1) |
| *Asterococcus superbus* (Cienk.) Scherff. | + | | | |
| *Auxenochlorella* sp. | | | | + (N = 1) |
| *Borodinellopsis oleifera* Schwarz | + | | | |
| *Borodinellopsis texensis* Dykstra [1] | | | + | |
| *Botryokoryne simplex* Reisig[1] | + | | | |
| *Bracteacoccus aggregatus* Tereg [1,2] | | + | | |
| *Bracteacoccus giganteus* H.W.Bischoff et H.C.Bold [1,2] | | + | + | |
| *Bracteacoccus grandis* H.W.Bischoff et H.C.Bold [1,2] | | | + | |
| *Bracteacoccus minor* (Schmidle ex Chodat) Petrová [1,2] | + | + | + | |
| *Bracteacoccus pseudominor* H.W. Bischoff et H.C. Bold [1,2] | + | | + | |
| *Bracteacoccus* spp. [1,2] | + | + | | + (N = 1) |
| *Carteria* sp. | | | + | |
| *Cecidochloris adnata* (Korshikov) H.Ettl | + | | | |
| *Chlamydocapsa* cf. *maxima* (Mainx) H.Ettl et Gärtner [1] | | | + | |
| *Chlamydocapsa lobata* Broady [1] | + | + | + | |
| *Chlamydocapsa* spp. | | + | | + (N = 1) |
| *Chlamydomonas* cf. *applanata* E.G.Pringsh. | | + | | |
| *Chlamydomonas* cf. *asymmetrica* Korshikov | | | + | |
| *Chlamydomonas elliptica* Korshikov | | | + | |
| *Chlamydomonas* cf. *gloeogama* Korshikov | | + | | |
| *Chlamydomonas hindakii* H.Ettl [1] | | + | | |
| *Chlamydomonas macrostellata* J.W.G.Lund [1] | | | + | |
| *Chlamydomonas* cf. *noctigama* Korshikov | | + | + | |
| *Chlamydomonas pseudagloë* Pascher | | | | + (N = 1) |
| *Chlamydomonas radiata* T.R.Deason et Bold | | | + | |
| *Chlamydomonas* cf. *reinhardtii* P.A.Dang. | | + | | |
| *Chlamydomonas* cf. *reisiglii* H.Ettl [1] | + | + | + | |
| *Chlamydomonas* cf. *thomassonii* H.Ettl | | + | | |
| *Chlamydomonas* spp. | + | + | + | + (N = 1) |
| *Chlorella chlorelloides* (Naumann) C.Bock, L.Krienitz et T.Pröschold | | | + | |

**Table 3.** *Cont.*

| Taxon | PU | SU | NU | SUm |
|---|:---:|:---:|:---:|:---:|
| **Chlorophyta** | | | | |
| *Chlorella vulgaris* W.Beij [1,2] | + | + | + | |
| *Chlorella vulgaris* W.Beij. f. *globosa* V.M.Andreeva [1,2] | + | + | | |
| *Chlorella* sp. | | | + | |
| *Chlorococcum costatozygotum* H.Ettl et Gärtner [1] | | | + | |
| *Chlorococcum infusionum* (Schrank) Menegh. | | + | | |
| *Chlorococcum isabeliense* P.A.Archibald et Bold [1] | | + | | |
| *Chlorococcum lobatum* (Korshikov) F.E.Fritsch et R.P.John [1] | | + | + | |
| *Chlorococcum* spp. | | + | + | |
| *Chloroidium lichinum* (Chodat) Darienko et Pröschold | | | | + (N = 1) |
| *Chloroidium* spp. | | | | + (N = 2) |
| *Chloroidium orientalis* Gontcharov, Abdullin, A.Nikulin, V.Nikulin et Bagmet | | | | + (N = 1) |
| *Chloroidium saccharophilum* (W.Krüger) Darienko, Gustavs, Mudimu, Menendez, Schumann, Karsten, Friedl et Proschold [1] | | | + | + (N = 1) |
| *Chlorolobion lunulatum* Hindák | | + | + | |
| *Chlorominima* sp. | | | | + (N = 1) |
| *Chloromonas reticulata* (Gorozh.) Gobi [1,2] [40] | | + | + | |
| *Chloromonas* cf. *rosae* H.Ettl. [1] | | | + | |
| *Chloromonas* spp. | | | | + (N = 2) |
| *Chloroplana terricola* Hollerb. | + | | | |
| *Chlororustica terrestris* (Herndon) Shin Watan., N.Mezaki et Tatsuya Suzuki | + | | | |
| *Chlorosarcinopsis* spp. | + | + | + | |
| *Chromochloris zofingiensis* (Dönz) Fucíková et L.A.Lewis | | | | + (N = 1) |
| *Coccomyxa confluens* (Kütz.) Fott | | | | + (N = 1) |
| *Coccomyxa elongata* Chodat et Jaag | | | | + (N = 1) |
| *Coccomyxa polymorpha* T.Darienko et T.Pröschold | | | | + (N = 1) |
| *Coccomyxa subellipsoidea* E.Acton | | | | + (N = 1) |
| *Coccomyxa subglobosa* Pascher | + | | + | |
| *Coccomyxa viridis* Chodat [1] | | + | | + (N = 1) |
| *Coccomyxa* spp. | | | | + (N = 13) |
| *Coelastrella aeroterrestrica* Tschaikner, Gärtner et Kofler | | | | + (N = 1) |
| *Coelastrella multistriata* (Trenkwalder) Kalina et Punčochárová | | | + | |
| *Coelastrella oocystiformis* (J.W.G.Lund) E.Hegewald et N.Hanagata [1,2] [42] | | + | + | + (N = 1) |
| *Coelastrella rubescens* (Vinatzer) Kaufnerová et Eliás [1,2] [42] | + | + | + | + (N = 1) |
| *Coelastrella terrestris* (Reisigl) E.Hegewald et N.Hanagata | + | | + | |
| *Coelastrella* spp. [1,2] [42] | | | + | + (N = 1) |
| *Coenochloris bilobata* (Broady) Hindák | + | | | |
| *Coenochloris signiensis* (Broady) Hindák [1] | + | + | + | |
| *Coenocystis* cf. *oleifera* (Broady) Hindák var. *antarctica* (Broady) V.M.Andreeva | | + | | |
| *Coenocystis* spp. | | | | + (N = 2) |
| *Coleochlamys apoda* Korshikov | + | | | |
| *Deasonia multinucleata* (Deason et H.C.Bold) H.Ettl et Komárek [1] | + | | + | |
| *Desmodesmus abundans* (Kirchner) E.Hegewald | | + | | |
| *Desmotetra stigmatica* (T.R.Deason) T.R.Deason et G.L.Floyd [1] | | | + | |
| *Dictyochloropsis splendida* Geitler | | | | + (N = 1) |
| *Dictyochloropsis* spp. | | | | + (N = 3) |
| cf. *Dictyococcus varians* Gerneck [1] | + | + | | |
| *Diplosphaera chodatii* Bialosukniá [1] | | + | | |
| *Diplosphaera chodatii* var. *mucosa* (Broady) Pröschold et Darienko | | + | | |
| *Elliptochloris bilobata* Tscherm.-Woess [1] | + | + | + | |
| *Elliptochloris reniformis* H.Ettl et G.Gärtner [1] | | + | + | |
| *Elliptochloris subsphaerica* (Reisigl) H.Ettl et G.Gärtner [1] | + | + | + | + (N = 1) |
| *Elliptochloris* spp. | + | + | | + (N = 6) |
| *Eremochloris* sp. | | | | + (N = 1) |
| *Ettlia minuta* (Arce et H.C.Bold) Komárek | + | | | |
| *Eubrownia aggregata* (R.M.Brown et Bold) Shin Watan. et L.A.Lewis | | + | + | |
| *Eubrownia dissociata* (R.M.Brown et Bold) Shin Watan. et L.A.Lewis | | + | + | |
| *Fernandinella semiglobosa* (F.E.Fritsch et R.P.John) Škaloud et Leliaert | | | + | |
| *Gloeococcus* sp. | | + | | |
| *Hariotina compacta* Wang, Liu, Hu et Liu | | | | + (N = 1) |
| *Herndonia botryoides* (Hernon) Shin Watan. | + | | | |
| *Heterochlamydomonas uralensis* Novakovskaya, Boldina, Shadrin, Patova [1,2] [59] | | + | | |
| cf. *Heterotetracystis intermedia* Ed.R.Cox et T.R.Deason | + | + | + | |
| *Interfilum paradoxum* Chodat et Topali | | + | | |
| *Interfilum terricola* (J.B.Petersen) Mikhailyuk, Sluiman, Massalski, Mudimu, Demchenko, Friedl et Kondratyuk | + | + | + | |
| *Keratococcus bicaudatus* (A.Braun ex Rabenh.) J.B.Petersen | | | + | |
| *Leptosira obovata* Vischer | | | | + (N = 1) |
| *Leptosira polychloris* Reisigl [1] | | + | + | |
| *Leptosira* spp. | | + | + | + (N = 1) |

**Table 3.** *Cont.*

| Taxon | PU | SU | NU | SUm |
|---|:---:|:---:|:---:|:---:|
| **Chlorophyta** | | | | |
| *Lobochlamys culleus* (H.Ettl) T.Pröschold, B.Marin, U.W.Schlösser et M.Melkonian | + | + | + | |
| *Lobosphaera incisa* (Reisigl) Karsten, Friedl, Schumannn, Hoyer et Lembcke [1] | | + | + | |
| *Macrochloris dissecta* Korshikov [1] | + | | + | |
| *Massjukichlorella* sp. | | | | + (N = 1) |
| *Microspora* sp. | | + | | |
| *Mychonastes homosphaera* (Skuja) Kalina et Punčoch. [1,2] | + | + | + | |
| *Myrmecia bisecta* Reisigl [1] | + | + | + | |
| *Myrmecia israelensis* (S.Chantanachat et H.Bold) T.Friedl | | | | + (N = 1) |
| *Myrmecia macronucleata* (Deason) V.M.Andreeva | + | | | |
| *Myrmecia pyriformis* J.B.Petersen | | | | + (N = 1) |
| *Myrmecia* sp. | | | | + (N = 1) |
| cf. *Nannochloris* sp. | | | + | |
| *Nautococcus terrestris* P.A.Archibald | + | | | |
| *Neochloris gelatinosa* Herndon | + | | | |
| *Neochloris* sp. | | + | | |
| *Neocystis broadiensis* Kostikov, Darienko, Lukesova et L.Hoffm. | | + | | |
| *Neocystis curvata* (Broady) Kostikov, Darienko, Lukesova et L.Hoffm. [1] | | + | + | |
| *Neocystis* sp. | | + | | |
| *Oedogonium* sp. | + | | | |
| *Palmellopsis gelatinosa* Korshikov | + | | | |
| *Parietochloris alveolaris* (H.C.Bold) Shin Watan. et G.L.Floyd [1] | + | + | + | |
| *Parietochloris bilobata* (Vinatzer) V.M.Andreeva [1] | + | + | + | + (N = 1) |
| *Parietochloris* cf. *pseudoalveolaris* (Deason et H.C.Bold) Shin Watan. et G.L.Floyd [1] | | + | | |
| *Parietochloris* spp. [1] | | + | | + (N = 1) |
| *Planophila asymmetrica* (Gerneck) Wille | | + | | |
| *Planophila laetevirens* Gerneck | | | | + (N = 1) |
| *Pleurastrum terricola* (Bristol) D.M.John [1] | + | + | + | + (N = 1) |
| *Pleurastrum* sp. | | | | + (N = 1) |
| *Pseudendoclonium* sp. | | + | | |
| *Pseudodictyochloris multinucleata* (Broady) H.Ettl et G.Gärtner | + | | | |
| *Pseudosphaerocystis* sp. | + | | | |
| *Pseudotrochiscia areolata* Vinatzer | + | | | |
| *Radiosphaera minuta* Herndon [1] | + | | + | |
| *Rhexinema* sp. | | | | + (N = 2) |
| *Sanguina nivaloides* Procházková, Leya et Nedbalová | | | | + (N = 1) |
| *Scenedesmus acuminatus* (Lagerheim) Chodat | | | | + (N = 1) |
| *Schizochlamydella minutissima* Broady [1] | | | + | |
| *Scotiellopsis levicostata* (Hollerb.) Punčoch. et Kalina | + | + | | |
| *Scotinosphaera* sp. | | | | + (N = 1) |
| *Spongiochloris lamellata* Deason et H.C.Bold | + | | | |
| *Sporotetras polydermatica* (Kütz.) Kostikov, Darienko, Lukesová et L.Hoffm [1] | + | + | + | |
| *Stichococcus bacillaris* Nägeli [1] | | + | + | |
| *Stichococcus* sp. | | + | | |
| *Stigeoclonium* sp. | | | | + (N = 1) |
| *Symbiochloris symbiontica* (Tscherm.-Woess) Skaloud, Friedl, A.Beck& Dal Grande | | | | + (N = 1) |
| *Tetracystis compacta* K.Schwarz [1] | + | | | |
| *Tetracystis excentrica* R.M.Brown et H.C.Bold | + | | | |
| *Tetracystis pulchra* R.M.Brown et H.C.Bold | + | | | |
| *Tetracystis tetraspora* (Arce et H.C.Bold) R.M.Brown et H.C.Bold [1] | | | + | |
| *Tetracystis* cf. *vinatzeri* H.Ettl et Gärtner. [1] | | | + | |
| *Tetracystis* spp. | | + | + | + (N = 1) |
| *Tetradesmus obliquus* (Turpin) M.J.Wynne | | + | | |
| *Tetrasporidium javanicum* Möbius | + | | | |
| *Trebouxia arboricola* Puymaly | + | | | |
| *Trebouxia* spp. | | + | + | |
| *Ulothrix tenerrima* Kutz. | + | + | + | |
| *Uvulifera mucosa* (Broady et M.Ingerfeld) Molinari | | | | + (N = 1) |
| *Uvulifera* sp. | | | | + (N = 1) |
| *Valeriella excentrica* (R.C.Starr) Darienko et Pröschold | + | | | |
| *Valeriella incrassata* (Chantanachat & H.C.Bold) Darienko et Pröschold | + | | | |
| *Watanabea reniformis* N.Hanagata, I.Karube, M.Chihara et P.C.Silva | | | | + (N = 1) |
| *Watanabea sichuanensis* Shuyin Li et al. | | | | + (N = 1) |
| *Xerochlorella* spp. | | | | + (N = 2) |
| Total | 58 | 68 | 67 | 80 OTUs Identified—78 Not identified—2 |
| **Charophyta** | | | | |
| *Closterium pusillum* Hantzsch [1] | + | + | | |
| *Cosmarium anceps* P.Lundell | | + | | |
| *Cosmarium saxicola* Kaiser | | + | + | |

**Table 3.** *Cont.*

| Taxon | PU | SU | NU | SUm |
|---|---|---|---|---|
| **Charophyta** | | | | |
| *Cosmarium undulatum* Corda ex Ralfs | | | + | |
| *Cosmarium* spp. | + | + | | |
| *Cylindrocystis brebissonii* var. *turgida* Schmidle | + | | + | |
| *Cylindrocystis crassa* De Bary | + | | + | |
| *Cylindrocystis* sp. | | + | | |
| *Euastrum binale* Ehrenb. ex Ralfs | + | + | | |
| *Klebsormidium crenulatum* (Kütz.) Lokhorst | | + | | |
| *Klebsormidium dissectum* (F.Gay) H.Ettl et G.Gärtner [1] | + | + | + | |
| *Klebsormidium flaccidum* (Kütz.) P.C.Silva, Mattox et W.H.Blackwel [1] | | + | + | |
| *Klebsormidium nitens* (Kütz.) Lokhorst [1] | + | + | + | + (N = 1) |
| *Mesotaenium caldariorum* (Lagerh.) Hansg. | | + | + | |
| *Mesotaenium endlicherianum* Nägeli | + | + | + | |
| *Mesotaenium pyrenoidosum* (P.A.Broady) Petlovany [1] | + | + | | |
| *Mesotaenium* sp. | | + | | |
| Total | 9 | 14 | 9 | 2 OTUs Identified—1 Not identified—1 |

Note: + taxon noted; [1]—strain isolated and deposited to SYKOA culture collection; [2]—molecular analyses for strain were performed; [40–42,58,59]—information about the strain is published in the source. The taxa within the phyla are in alphabetical order.

The distribution of taxa at different sampling sites is presented in Figure 2. The genera *Chlamydomonas*, *Leptolyngbya*, *Eunotia*, *Chlorococcum*, *Gloeocapsa*, *Pinnularia*, *Nostoc*, *Aphanothece*, *Bracteacoccus*, and *Phormidium* contributed significantly to the species diversity, with the largest number of species among the studied taxa (Figure 3a). Direct microscopy and cultural methods showed that natural BSC samples, on average, contain from 15 to 50 species of cyanobacteria and eukaryotic algae [27].

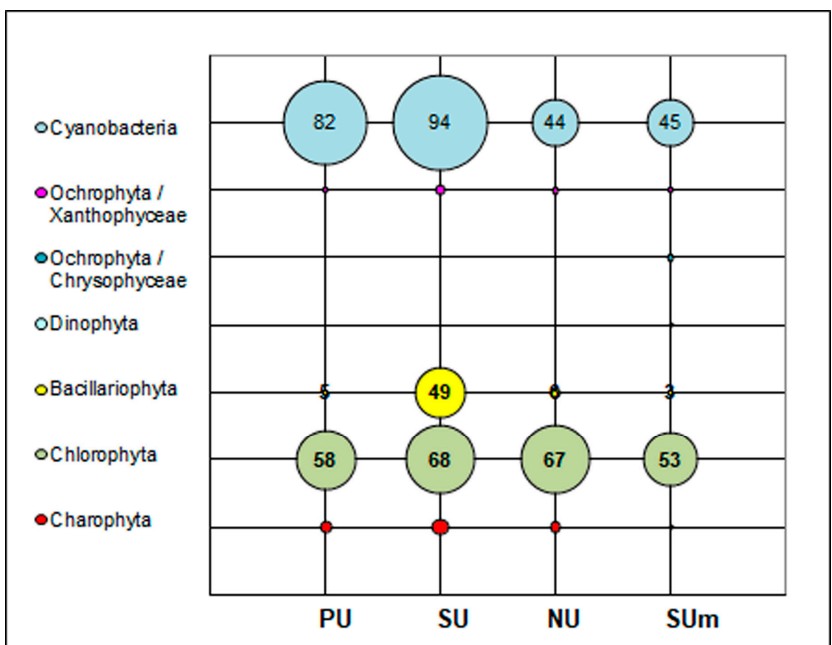

**Figure 2.** Distribution of diversity (in circles are the number of taxonomic units below the genera) of cyanobacteria and algae at phylum level in BSCs in the northern regions of the Urals. The abbreviations of the sample sites are in Table 1.

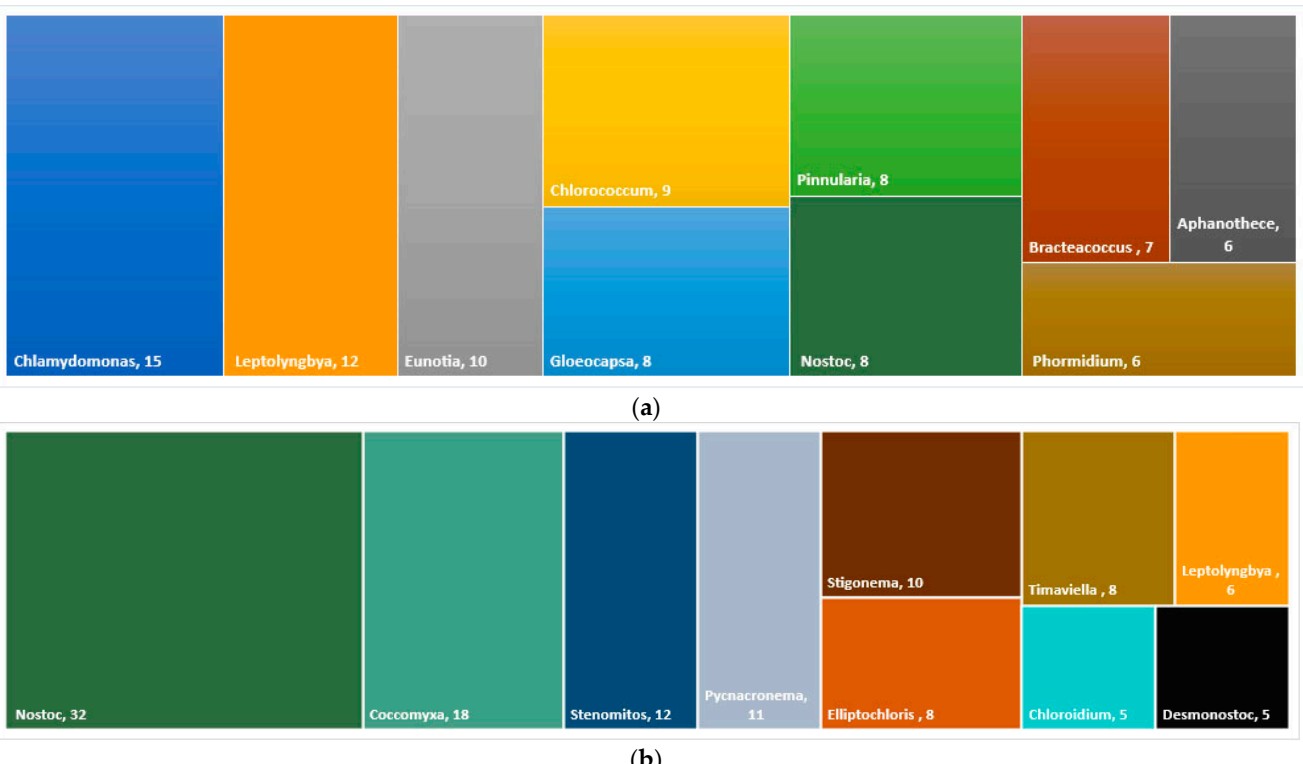

**Figure 3.** Leading genera of algae and cyanobacteria in BSCs of the Urals in terms of diversity based on: (**a**)—morphological (by number of taxa) and (**b**)—metabarcoding (by number of OTUs) analysis. Matching genera marked with the same color.

### 3.2. Diversity of Algae and Cyanobacteria in BSCs Based on Metabarcoding Data

A total of 351 OTUs were assigned to soil algae (130) and cyanobacteria (221) through metabarcoding, from which only 135 cyanobacteria OTUs and 100 OTUs of eukaryotes were identified at a species or genus level (Table 3 and Supplementary File S1). The remaining OTUs had unclear taxonomic states and were excluded from further analyses in this study. Among the cyanobacteria, 13 were identified at the species level and 122 according to their genus rank (Table 3). OTUs at the genus rank may represent either new species or species for which there is no molecular data deposited in reference databases. In our study, 47 taxa, revealed through a morphological survey (Table 3), had missed nucleotide sequences in the NCBI database (https://www.ncbi.nlm.nih.gov/, accessed on 29 August 2023). Nine cyanobacteria species were discovered for the first time in the BSCs of the Urals regions only through metabarcoding data. These species are cryptic, with similar morphology to the described taxa, and their identification without molecular data is considered challenging. Chlorophyta had 80 OTUs, of which 44 were identified only by genus rank and 34 by species rank (23 species new to the Ural region) (Table 3). Moreover, 11 OTUs were assigned to Bacillariophyta, of which four genera and one species (new to the region) were identified. Ochrophyta had 23 OTUs; 6 were identified by species rank (5 new to the region) and 8 by genus rank. Dinophyta had 8 OTUs, and only 2 were identified to a genus rank. Cryptophyta and Euglenophyta had 3 OTUs per each group, but they were not identified. Charophyta had 2 OTUs; one was identified by species rank (Table 3).

Genera *Nostoc*, *Coccomyxa*, *Stenomitos*, *Pycnacronema*, *Stigonema*, *Elliptochloris*, *Timaviella*, *Leptolyngbya*, *Chloroidium*, and *Desmonostoc* were the most diverse in terms of the number of taxonomic units identified both among cyanobacteria and eukaryotes (Figure 3b).

The number of OTUs varied between different types of BSCs and ranged from 25 to 129 per sample, from which eukaryotes accounted for 6 to 33 OTUs and cyanobacteria from 15 to 96 OTUs.

According to morphological data, there are 55 common species in BSCs in all the studied regions (Figure 4a). According to morphological and metabarcoding data only 23 common taxa between SU and SUm regions (Figure 4b). The distribution of the identified diversity of algae and cyanobacteria for different regions in the Urals was not the same.

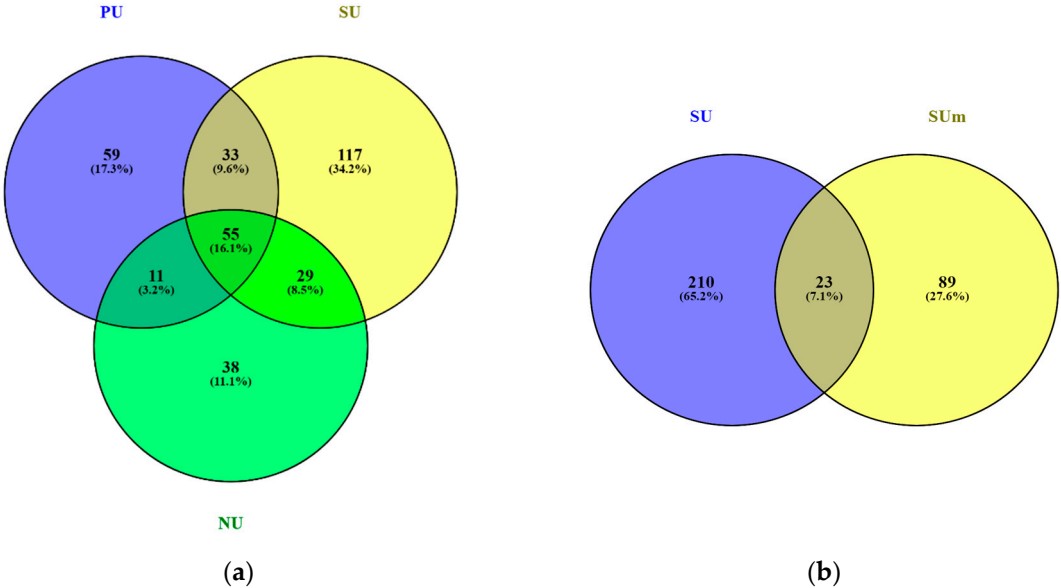

**Figure 4.** Venn diagram analysis (https://bioinfogp.cnb.csic.es/tools/venny/, accessed on 20 August 2023) of BSCs diversity (rank below genus) in different regions of the Urals. (**a**)—according to morphological data for PU, SU, and NU; (**b**)—according to morphological data for SU and metabarcoding data for SUm.

## 4. Discussion

The combination of the morphological approach with metabarcoding data provided a more comprehensive view of the current diversity of soil algae and cyanobacteria in pioneer vegetation communities of mountain tundra in the northern regions of the Urals. The diversity of the BSCs in the region was found to consist of 422 taxa (343 taxa—morphological data and 235 taxa—metabarcoding data) of soil algae and cyanobacteria (Table 3). The use of DNA metabarcoding provided complementary data, revealing the 29 cyanobacteria taxa and 42 eukaryotic algae taxa not described before in the north Ural regions. In addition, 86 OTUs of cyanobacteria and 30 OTUs of eukaryotic algae remained unidentified by species or genus rank. Our study's findings suggest a potential species pool for BSCs in the mountain tundra of the Urals can reach more than 500 species. According to W. Büdel et al. [10], there are more than 320 species of cyanobacteria from 70 genera found in BSCs worldwide. Our results are consistent with the literature data on BSCs, which shows the diversity of soil algae and cyanobacteria in the Arctic [4,7–9,15,16] and mountainous regions [2,19,28,60].

The BSCs contain a variety of algal and cyanobacterial species that are rare and difficult to identify based on morphological approaches alone [52,61,62]. The DNA metabarcoding method not only confirmed the known diversity of algae groups in BSCs and typical species but also allowed for the identification of new taxonomic groups at various levels of classification, from phylum to species. This method also helped to detect hidden diversity, for example, by revealing cryptic species, taxonomically challenging species, or novel species outside their known habitat [18,45]. Specifically, the metabarcoding analysis identified algae from Dinophyta (2 taxa) and Chrysophyceae (5 taxa), and these were recorded for the first time in the soils of the northern regions of the Urals (Table 3). Chrysophyceae algae have been previously detected in BSCs from polar regions using metabarcoding [18]. Some of the identified species of cyanobacteria and algae were found to be reported outside

their typical habitat and geographic distribution. For example, species from the genera *Hariotina* and *Stigeoclonium* were previously only reported in aquatic environments [43]. The cyanobacteria *Dactylothamnos antarcticus* were described to be in the littoral of streams and lakes and on wetted rocks in the Antarctic [13]. For example, a cryptic cyanobacteria species, *Oculatella crustae-formantes*, discovered in BSCs of the Urals was found in the BSCs of the Arctic Spitsbergen, Norway [62]. The green algae species in the cryptic genus *Parietochloris* [63] were detected in the studied BSCs (Table 3). Similarly, one species from the genus *Planophila* was revealed in the BSC of the Urals only through metabarcoding, since representatives of this genus have high phenotypic plasticity [64]. Moreover, metabarcoding helped to identify lichen photobionts from the genus *Asterochloris* (Table 3), which is a significant proportion of the BSCs [65]. The species from the genus *Leptosira*, revealed in the study, also characterize high hidden species diversity and high morphological variability [66]. These findings challenge the traditional concepts of some algal species distribution and highlight the importance of exploring the potential diversity of organisms in various geographic locations. Combining these two approaches can provide a more complete picture of species diversity and improve our understanding of ecosystem functioning.

Cyanobacteria and Chlorophyta had the highest number of taxonomic units, confirmed by both morphological observations and metabarcoding (Figure 2). Their dominance in this diversity study is not surprising, as these phyla are commonly found in BSCs [7,10,21,60]. The genera *Nostoc, Coccomyxa, Chlamydomonas, Leptolyngbya, Stenomitos, Pycnacronema, Stigonema*, and *Eunotia* were the most diverse in terms of the number of taxonomic units identified (Table 3), suggesting that these genera may play an important role in the ecology of the BSCs in the studied region [20–26] and in BSCs of other northern and mountainous regions [7,10,18]. The compositions of the leading genera obtained through cultural and metabarcoding methods have low similarity (Figure 3). This is likely due to the limited data obtained from the metabarcoding tool and the detection of a high number of species that are not distinguishable through cultural methods. It was earlier reported that BSCs are often dominated by poorly cultivated species in terms of quantitative measures [7,10].

Previous study using light and fluorescent microscopy also have shown that cyanobacteria and green algae form the phototrophic basis of BSCs, with densities ranging from 5.75 to 27.10 million cells/g of soil [27]. The study identified the dominant species in BSCs, including species from Cyanobacteria (such as *Nostoc commune* (Figure 5a), *Stigonema minutum* (Figure 5b), *S. ocellatum, Gloeocapsopsis magma, Scytonema hoffmannii* (Figure 5c), and species from the genera *Aphanocapsa, Fischerella, Leptolyngbya, Phormidium*, and *Schizothrix*) and Chlorophyta species (such as *Chlamydocapsa lobata, Elliptochloris bilobata* (Figure 5d), *E. subsphaerica, Coccomyxa simplex, Sporotetras polydermatica* (Figure 5e), *Lobochlamys culleus* (Figure 5f), and species from the genera *Coelastrella, Cylindrocystis, Leptosira, Myrmecia* and *Mesotaenium*). The listed species are typical inhabitants of BSCs in mountainous and polar regions and are highly resistant to a prolonged lack of moisture and sudden changes in soil temperature in the upper horizons, likely due to their small size, thickened cell wall, rapid reproduction, and their ability to form slimy colonies.

The alpha diversity of cyanobacteria and algae in BSCs ranged from 133 to 234 species in different regions of the Urals (Figures 2 and 4; Table 3). The number of common species among all the investigated regions was 55. The highest number of species was found in the BSCs of Subpolar Urals, while the Polar Urals had the second highest species diversity. The fewest species were identified in the Northern Urals, where algae research is just beginning [22]. This suggests that different types of BSCs in different regions of the Urals have specific groups of cyanobacteria and algae, and the overall diversity of phototrophs in BSCs is not high. Among the 55 species recorded in all regions of the Northen Urals, the following 13 taxa were confirmed by metabarcoding data: cyanobacteria—*Microcoleus vaginatus, Nostoc commune, N. edaphicum*, and *Tolypothrix distorta*; and eukaryotic algae—*Chloroidium saccharophilum, Coccomyxa viridis, Coelastrella oocystiformis, C. rubescens, Elliptochloris subsphaerica, Klebsormidium nitens, Parietochloris bilobate, Pleurastrum terricola*, and *Vischeria magna*. Almost all of these species belong to the dominant complex or have a

high occurrence in the biological soil crusts of the diverse mountain tundra communities of the Polar, Subpolar, and Northern Urals [20–27].

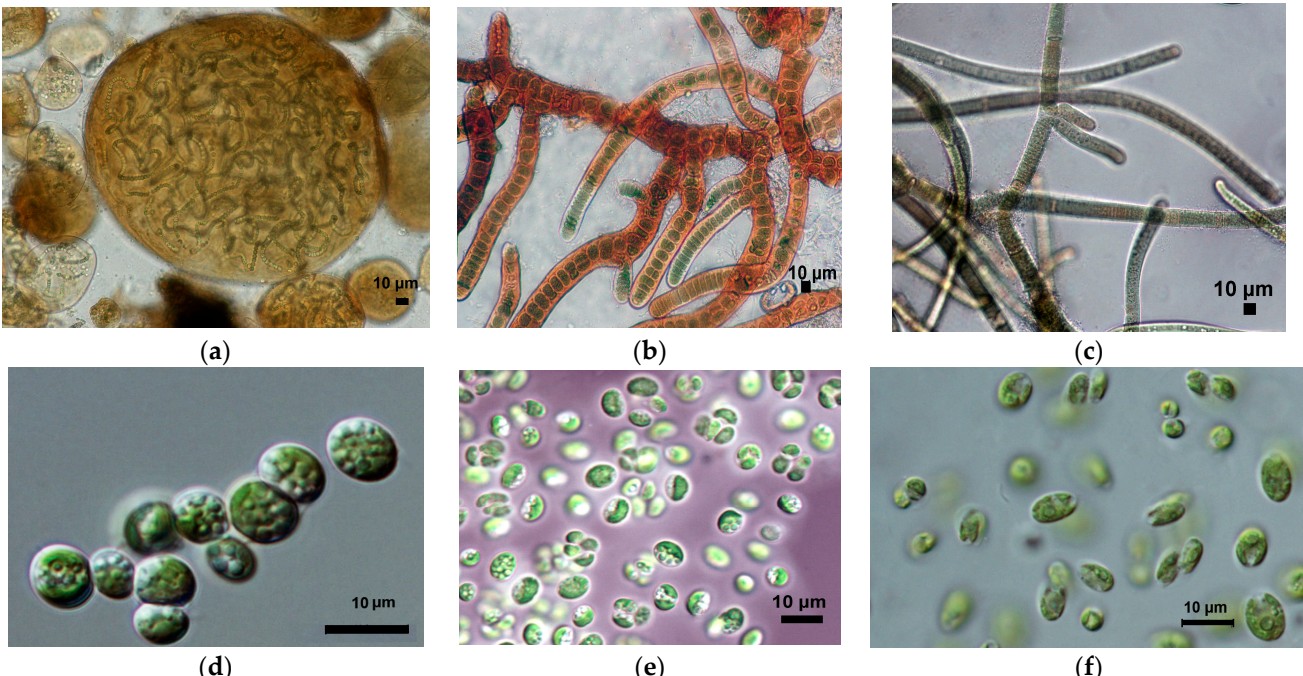

**Figure 5.** Cyanobacteria and green algae dominated in biological soil crusts of spotted mountain tundra: (**a**)—*Nostoc commune*, (**b**)—*Stigonema minutum*, (**c**)—*Scytonema hoffmannii*, (**d**)—*Elliptochloris bilobata*, (**e**)—*Sporotetras polydermatica*, and (**f**)—*Lobochlamys culleus*.

Long-term studies have shown that on average, 15–50 species of cyanobacteria and eukaryotic algae are typically found in BSCs per site in the Northern Urals [22,25–27]. However, according to the literature data, the diversity of algae in BSCs can highly vary depending on the region, with the number of taxa typically ranging from 18 to 65 [19,21,30,60]. Molecular data have shown that the diversity in one BSC can reach large values (from 25 to 129 taxa per sample).

## 5. Conclusions

Our results show that BSCs are unique microecosystems in mountain environments with a high biodiversity of phototrophic components. A morphological approach identified 343 taxa of terrestrial cyanobacteria and eukaryotic algae in BSCs from various mountain communities in the northern regions of the Urals, and using metabarcoding, 235 taxa were identified. Metabarcoding studies have supplemented the understanding of the hidden diversity of the BSC phototrophic community by 48 taxa of the generic and 38 species levels. Both methods produce data comparable in terms of the number of species per sample, which makes it possible to use the metabarcoding method for a correct assessment of diversity and monitoring. The discovered dominant groups, consisting of cyanobacteria and green algae, are also found to be the main phototrophic part of BSCs in other regions of the Earth. Overall, the study found that the diversity of algae and cyanobacteria in BSCs varies across the regions of the Urals. The study contributes to our knowledge of the passive and active biodiversity of algae and cyanobacteria in BSCs and their distribution in mountain ecosystems.

This research focuses primarily on organisms that are sufficiently well studied to enable the correct identification of genus and species from morphological and/or molecular data. However, a full analysis of the metabarcoding data reveals a very significant part of the latent diversity and reveals a common problem in the interpretation of these data at the

moment. OTUs identified with unclear taxonomic positions may be new lineages, and it is unclear whether they are photosynthetic or heterotrophic organisms. These data may also include identification errors due to incomplete reference databases. One thing is certain, that further improvement to the approaches used for their study, both morphological and molecular, is necessary to more fully assess the diversity of these specific habitats.

Further studies of BSCs should aim to identify the interaction of organisms with each other, the seasonal development of BSCs, the study of various functional groups, as well as long-term succession under climate change. This is only possible with a combination of different approaches to the study of these key components of mountain and arctic ecosystems. The use of metabarcoding to study terrestrial algae and cyanobacteria will not only provide new information about their species and functional diversity but will also change our understanding of their ecology and distribution in the BSCs of the different climatic zones of the Earth. However, the successful application of metabarcoding is limited by the obvious incompleteness of the databases of the nucleotide sequences of algae and cyanobacteria. Therefore, to develop this approach, more effort is required to isolate algae and cyanobacteria into culture from different climatic zones and their taxonomic revision.

**Supplementary Materials:** The following supporting information can be downloaded at https://www.mdpi.com/article/10.3390/d15101080/s1, File S1: The sequences of identified OTUs of cyanobacteria and eukaryotic algae.

**Author Contributions:** Conceptualization, E.P., E.G. and I.N.; methodology, E.P., E.G. and I.N.; bioinformatic analysis, E.G. and N.M.; field investigation, E.P. and I.N.; data curation, writing—original draft preparation, writing—review and editing, visualization—all authors; project administration and funding acquisition, E.P. All authors have read and agreed to the published version of the manuscript.

**Funding:** This research was funded by the Russian Science Foundation, grant number: №22-24-00673 (https://rscf.ru/project/22-24-00673/).

**Institutional Review Board Statement:** Not applicable.

**Data Availability Statement:** The data presented in this study are available on request from the corresponding author.

**Conflicts of Interest:** The authors declare no conflict of interest. The funders had no role in the design of the study; in the collection, analyses, or interpretation of data; in the writing of the manuscript; or in the decision to publish the results.

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
