# Peer review of "Diversity of Cyanobacteria and Algae in Biological Soil Crusts of the Northern Ural Mountain Region Assessed through Morphological and Metabarcoding Approaches"

_diversity, doi:10.3390/d15101080_

Round 1
Reviewer 1 Report
The manuscript investigates a Diversity of cyanobacteria and algae in biological soil crusts of the
northern Ural mountainous regions assessed through morphological and metabarcoding approaches. A good study has been conducted, but this article needs more edits, and some points should be addressed before its acceptance in Diversity Journal.
1. Please revise the manuscript for English proofreading carefully throughout all manuscript. I see unnecessary full stop and capital letters.
2. The introduction is very brief, and requires more background information.
3. I suggest adding Table 3 as a supplementary Table. And or using graphics to present Tables 3 would be more effective.
4. The title includes “morphological”. However, there are no morphological micrographs of any samples in the text.
5. Line 160-161, The quality control of the obtained pools of libraries was carried out using the Fragment Analyzer system, quantitative analysis was carried out using qPCR. Where was the qPCR method used in the results and what results were obtained. What are the primers and conditions of qPCR, please add clearly.
6. ln 2.5. Amplification, library preparation and sequencing. How was the sample treated? And how many replicates were made? Was there statistical analysis and repeatability test.
7. The discussion should be based on the obtained results, and remove overly extended content.
8. In Conclusion,what are the gaps in this manuscript and the recommended studies in the future?
Please revise the manuscript for English proofreading carefully throughout all manuscript, including highlights, figures, and tables. Also, I see very long sentences. Try to shorten them.
Author Response
Dear colleges,
We wanted to thank the reviewers for their valuable feedback. We carefully addressed their suggestions in our revised manuscript. We believe that this noticeably improved the clarity of our manuscript and now it meets the expectations to be published. We appreciate the time and expertise of the reviewers and the editor to guide us through this process!
We would like to note that the datasets analysed during the current study are available in the NCBI repository (accession: PRJNA1016378).
Sincerely,
Patova Elena and coauthors.
FOR REVIEWER 1.
We want to thank you for your helpful comments on our manuscript. We've made the changes you suggested. And below we address each comment separately.
- Please revise the manuscript for English proofreading carefully throughout all manuscripts. I see unnecessary full stops and capital letters.
Answer: We checked the language once again. All changes are coloured in the text. The final version of the manuscript, including all changes, was checked by a translator.
- The introduction is very brief and requires more background information.
- The discussion should be based on the obtained results, and remove overly extended content.
Answer: We've added some extra text to the introduction and tried to shorten the discussion, as you recommended.
- I suggest adding Table 3 as a supplementary Table. Using graphics to present Table 3 would be more effective.
Answer: Table 3 summarises the main results of our work. While we understand that the taxonomic data presented is extensive, we believe that such lists are crucial for future studies and comparisons. It could be quite valuable for researchers working in similar habitats but at different geographical locations, such as the Arctic, Antarctic, mountains, polar regions, and arid deserts. Additionally, information on species composition is vital for aiding and improving the interpretation of metagenomic data in BSCs.
- The title includes “morphological”. However, there are no morphological micrographs of any samples in the text.
Answer. We've added micrographs (Figure 5) of dominant cyanobacteria and green algae species to the manuscript.
- Line 160-161, The quality control of the obtained pools of libraries was carried out using the Fragment Analyzer system, quantitative analysis was carried out using qPCR. Where was the qPCR method used in the results and what results were obtained. What are the primers and conditions of qPCR, please add them clearly.
Answer: We want to clarify that we did not use the qPCR method to obtain our results. It was a quality control check by a sequencing company to check if the obtained pool of libraries had a sufficient quantity and quality for sequencing.
- ln 2.5. Amplification, library preparation and sequencing. How was the sample treated? And how many replicates were made? Was there statistical analysis and repeatability test?
Answer: We have used three biological replicates and a “mock” community (mock community - in situ positive control for amplicon sequencing of algae and cyanobacteria from the same ecosystem) as a control to monitor success of sequencing with selected primers.
- In Conclusion, what are the gaps in this manuscript and the recommended studies in the future?
Answer: We added further text into a conclusion:
In conclusion, future studies of BSCs should focus on understanding organism interactions, seasonal development, various functional groups, and long-term succession, especially, in the context of climate change. These goals require a combination of various research approaches, both traditional and novel. A metabarcoding to study terrestrial algae and cyanobacteria offers significant potential to enhance our understanding of diversity and ecological roles across different climatic zones. However, this approach is limited by gaps and a lack of sequences for these organisms in databases. To further develop metabarcoding, efforts are needed to culture and taxonomically revise algae and cyanobacteria from various climatic zones.

Reviewer 2 Report
This study ivestigated cyanobacterial and eukaryotic algal composition in biological soil crusts (BSCs) of the northern Ural mountainous regions using morphological and metabarcoding approaches. Authors showed that 343 taxa were identified by the morphological approach and 351 OTUs were obtained by the pyrosequencing method, and they concluded that total 422 taxa were found by comination of both methods. Basically algal diversity in BSCs system is very important and interesting issue for biodiversity study, and this study provided some important source and data collection to science, and high algal diversity shown in this study will also emphasize the importance for studying algal diversity in the BSCs systems worldwide. I recommended this manuscript to be published but need to be considerably revised before acceptance as following suggestions:
1. Table 3 is too big, and it contained too much contents, and I suggest it to be futher divided, and cultivated strains should be seperately listed, and readers can easily see which strains and which species have been isolated and cultivated. The phylogenetic tree based on these cultivated strains should be provided.
2. 422 taxa was simply got by adding the numbers of OTUs and microscopy observation, and this is not correct. They are parallel approaches, and the data obtained by these two methods can be compared, but can not added to get a total number of taxa for revealing diversity.
3. There is no detailed explanation for figure 3, and I don't know the meanings of colors and squares in both a and b, and there were even some Russian words after genus names.
Author Response
ID diversity-2576658
"Diversity of cyanobacteria and algae in biological soil crusts of the northern Ural mountainous region assessed through morphological and metabarcoding approaches"
Diversity
Dear colleges,
We wanted to thank the reviewers for their valuable feedback. We carefully addressed their suggestions in our revised manuscript. We believe that this noticeably improved the clarity of our manuscript and now it meets the expectations to be published. We appreciate the time and expertise of the reviewers and the editor to guide us through this process!
We would like to note that the datasets analysed during the current study are available in the NCBI repository (accession: PRJNA1016378).
Sincerely, Patova Elena and coauthors.
FOR REVIEWER 2.
We are grateful to you for your review of our manuscript and valuable comments. We added recommended changes to the manuscript.
- Table 3 is too big, and it contained too much contents, and I suggest it to be futher divided, and cultivated strains should be seperately listed, and readers can easily see which strains and which species have been isolated and cultivated. The phylogenetic tree based on these cultivated strains should be provided.
Answer: Table 3 presents the main results of our work. We agree that taxonomic data is always large, but dividing the list of species into cultivated and non-cultivable strains creates difficulties for the analysis of the material. In addition, there are not many cultivated strains, and they are marked with a special mark in Table 3.
We believe that for this work, constructing general phylogenetic trees will not be indicative. Phylogenetic trees are usually used to revise algal groups in special taxonomic works and include different sets of genetic markers for each group. Constructing a single tree for even one group, such as green algae or cyanobacteria, is not possible or will be uninformative if based on a single conserved marker, such as the widely used 18S (for eukaryotes) and 16S (for prokaryotes). Therefore, we supplemented Table 3 for cultivated strains with links to our publications, where information about their phylogenetic analysis can be found.
- 422 taxa was simply got by adding the numbers of OTUs and microscopy observation, and this is not correct. They are parallel approaches, and the data obtained by these two methods can be compared, but can not added to get a total number of taxa for revealing diversity.
Answer: The main goal of this study is to summarize data on the diversity of BSCs from different regions of the Urals, obtained using different approaches. Therefore, we believe that summing up the results of 2 methods (morphological and molecular genetic) is justified. In addition, in Table 3, species identified using the metabarcoding method are separated from other taxa and are easy to see. Of course, each method has certain limitations. We discuss this in detail in the text and provide a separate comparison of the two methods. Changes were made to the conclusions to address this.
- There is no detailed explanation for Figure 3, and I don't know the meanings of colors and squares in both a and b, and there were even some Russian words after genus names.
Answer: Thank you for your comment. We changed the format of the pictures to jpg (it might have been a connected software issue) and adjusted the colour of the squares. Matching genera for morphological and metabarcoding data are now marked with the same color.

Round 2
Reviewer 1 Report
The authors have addressed most of the comments. There is no need to launch another round of review process. I agree to the article being accepted for publication in this journal.